# Exploring the Association between Urine Caffeine Metabolites and Urine Flow Rate: A Cross-Sectional Study

**DOI:** 10.3390/nu12092803

**Published:** 2020-09-13

**Authors:** Shou En Wu, Wei-Liang Chen

**Affiliations:** 1Division of Family Medicine, Department of Family and Community Medicine, Tri-Service General Hospital and School of Medicine, National Defense Medical Center, Taipei 114, Taiwan; grace830115@gmail.com; 2Division of Geriatric Medicine, Department of Family and Community Medicine, Tri-Service General Hospital, and School of Medicine, National Defense Medical Center, Taipei 114, Taiwan; 3Department of Biochemistry, National Defense Medical Center, Taipei 114, Taiwan

**Keywords:** urine caffeine metabolites, urine flow rate

## Abstract

Examination of urine excretion of caffeine metabolites has been a simple but common way to determine the metabolism and effect of caffeine, but the relationship between urinary metabolites and urine flow rate is less discussed. To explore the association between urinary caffeine metabolite levels and urine flow rate, 1571 participants from the National Health and Nutrition Examination Survey (NHANES) 2011–2012 were enrolled in this study. We examined the association between urinary caffeine metabolites and urine flow rate with linear regression models. Separate models were constructed for males and females and for participants aged <60 and ≥60 years old. A positive association was found between concentrations of several urinary caffeine metabolites and urine flow rate. Three main metabolites, namely, paraxanthine, theobromine, and caffeine, showed significance across all subgroups. The number of caffeine metabolites that revealed flow-dependency was greater in males than in females and was also greater in the young than in the elderly. Nevertheless, the general weakness of NHANES data, a cross-sectional study, is that the collection is made at one single time point rather than a long-term study. In summary, urinary concentrations of several caffeine metabolites showed a positive relationship with the urine flow rate. The trend is more noticeable in males and in young subgroups.

## 1. Introduction

Caffeine is a common psychoactive stimulant that can be found in daily beverages such as coffee, tea, and cocoa. Its impact on various aspects has aroused the interest of researchers from different fields. In the field of physiology, the impact of caffeine on the central nervous system and peripheral organs has been widely discussed [1]. The most noticeable one is the antagonism of adenosine receptors A1 and A2 [2], which play a famous role in arousal, vigilance, and anxiety [3]. In regard to neurotransmitters, caffeine seems to affect norepinephrine, dopamine, and serotonin, which contribute to alertness [4]. In the cardiovascular system, caffeine increases heart rate and affects blood pressure, cardiac rhythm, and various cardiac diseases [5,6]. In kidneys, caffeine induces diuresis and natriuresis [7]. In the field of psychology, sleep disturbance, learning and memory, addiction, and withdrawal are popular topics related to caffeine [8,9,10]. 

In the field of pharmacology, enzyme assays, including CYP1A2, N-acetyltransferase, and xanthine oxidase, utilize caffeine and its urinary metabolites as the means of evaluation [11,12]. There are various ways of evaluating caffeine metabolism. Urine levels, serum levels, and metabolite ratios of caffeine act as biomarkers for diseases [13], targets for drugs [14], and probes for enzyme activity [15]. Factors that may confound the results of examinations should be taken into account when interpreting related data. Urine flow rate is undoubtedly a crucial factor when interpreting data regarding urinary caffeine metabolism, and thus the association between urinary caffeine metabolite concentrations and urine flow rate deserves attention. Previous literature discussing flow-dependency put more focus on theophylline, one of the caffeine metabolites that is well known for its therapeutic effects on asthma and chronic obstructive pulmonary disease (COPD) [16,17,18]. However, comprehensive studies about other caffeine metabolites are lacking. Therefore, the purpose of our study is to investigate the relationship between 14 main urinary caffeine metabolites and the urine flow rate.

## 2. Materials and Methods 

### 2.1. Design and Participants

The NHANES study, a nationally representative study of population in the United States, is a cross-sectional survey based on a national sample of randomly-selected residents in the USA. It is administered by the National Center for Health Statistics (NCHS) of the Centers for Disease Control and Prevention (CDC). The survey combines three main parts. Initial screening questions determine qualified participants. Afterward, an extensive interview is held at home, which includes information such as age, gender, race, medical history, and health status. Further physical examination or clinical evaluations are performed at specially designed mobile examination centers (MECs). All interviewers received training programs and met the required standards. NHANES started in 1999 and remains a continuous annual survey, with data released every 2 years. Detailed questionnaire instruments, procedure manuals, brochures, and consent documents for the 2011–2012 NHANES are described on the NHANES website. This study gained Institutional Review Board (IRB; project identification code protocol #2011-17) approval by the National Center for Health Statistics (NCHS) in line with the revised Helsinki Declaration. Informed consents were collected from all research participants before the data-gathering procedure and examinations were carried out.

There were 9756 participants in the NHANES dataset from 2011–2012. Data from 2009–2010 also performed urinary caffeine analysis but was abandoned due to the instrument used not being suitable for analyzing both positive and negative ions simultaneously, so each urine specimen was analyzed twice. Data from 2011–2012 was from an improved instrument so that the measurement of each specimen was done in a single analysis. After excluding those under 18 years old and those with missing data such as urine flow rate, urinary caffeine analysis, and those taking medication of benign prostatic hyperplasia and diuretics, 1410 eligible participants were involved in our analysis. Figure 1 shows a scheme of the flow chart of participant recruitment. We performed our analyses in three stages: categorizing participants as a whole population, by gender (male and female), and by age (cutoff value set at 60 years old to refer to the elderly population [19,20]).

### 2.2. Measurement of Caffeine Metabolites in Urine

Spot urine samples were collected by experienced operators at the MECs. Recorded documents included the date and time of sampling and the volume of urine collection. Samples were stored at ≤−70 °C based on the Laboratory Procedures Manual before transportation to the National Center for Environmental Health (Centers for Disease Control and Prevention, Atlanta, GA, USA) for testing. Urinary metabolite quantification was determined by ultra-high performance liquid chromatography–electrospray ionization–tandem quadrupole mass spectrometry (UHPLC–ESI–MS/MS) (Agilent Technologies, Palo Alta, CA, USA) with stable isotope-labeled internal standards. More detailed methods are reported on the NHANES website.

Caffeine and 14 of its urinary metabolites, 15 in total, were examined, including 1-methyluric acid, 3-methyluric acid, 7-methyluric acid, 1,3-dimethyluric acid, 1,7-dimethyluric acid, 3,7-dimethyluric acid, 1,3,7-trimethyluric acid, 1-methylxanthine, 3-methylxanthine, 7-methylxanthine, 1,3-dimethylxanthine (theophylline), 1,7-dimethylxanthine (paraxanthine), 3,7-dimethylxanthine (theobromine), 1,3,7-trimethylxanthine (caffeine), and 5-acetylamino-6-amino-3-methyluracil (AAMU). AAMU is the decomposition product of the relatively unstable caffeine metabolite 5-acetylamino-6-formylamino-3-methyluracil (AFMU). Samples were allowed to incubate for at least 30 min at room temperature so that conversion of all AFMU to the more stable AAMU was ensured.

The lower limit of detection (LLOD in umol/L) for caffeine and caffeine metabolites can be obtained from the NHANES website. For analytes with results below the lower limit of detection, the value is the lower limit of detection divided by the square root of 2 (LLOD/sqrt [2]). All presented data satisfied quality control (QC) procedures, which were performed by a multirule quality control system. Samples examined were collected from 3 QC pools (low-, medium-, and high-quality control pools). Urine analyte concentrations were adjusted to urinary creatinine (uCr) by dividing urine concentration of metabolites by uCr values.

### 2.3. Measurement of Urine Flow Rate

Urinary flow rate was not a regular examined item in every cycle of NHANES. We collected our data from NHANES 2011–2012. Upon visiting MECs, participants reported the time of the last urinary void at home. At the center, the urinary volume was measured, and the time of sample collection was recorded. Up to three voids could be collected if the initial two voiding volumes were insufficient for the clinical and laboratory analyses. Conceptually, the calculation of urine flow rate is by dividing the volume of the present urine sample by the time duration between the former urination and the present urine collection, i.e., (total urine volume)/(total time duration).

### 2.4. Covariates

The self-reported demographic details of all subjects comprise gender, age, race/ethnicity, smoking history, and medical history. Race was sorted into groups including Mexican American, other Hispanic, non-Hispanic whites and blacks, and other races. Both former and current smokers were defined as having a habit of smoking. The formula of body mass index (BMI) is weight in kilograms divided by height in meters squared (kg/m^2^). Heart disease was defined as ever been diagnosed with congestive heart failure, coronary heart disease, angina, or heart attack. Biochemical data are measured as follows: aspartate aminotransferase (AST) were detected by the Beckman Coulter UniCel DxC 800 Synchron Clinical System; fasting plasma glucose (FPG) levels and urine creatinine (Cr) were measured by Roche/Hitachi Modular P Chemistry Analyzer. Further details about collection procedures are available on the NHANES website.

### 2.5. Statistical Analysis

We conducted a statistical analysis using SPSS (IBM SPSS Statistics for Windows, version 22.0, released 2013; IBM Corp., Armonk, NY). Qualitative data and quantitative variables were reported in percentages and medians and interquartile ranges (IQRs), respectively. A *p*-value of ≤0.05 was considered statistically significant. The urine flow rates deviated from normality, and, thus, log-transformation was performed to achieve normalization. Subsequently, we applied linear regression models to investigate the relationship between urine levels of caffeine metabolite levels and the log-transformed urine flow rate. 

Four models were provided in each analytic group to adjust for relevant covariates. The unadjusted model was numbered Model 1; Model 2 was adjusted for age, gender, and race; Model 3 was further adjusted for BMI, serum fasting glucose, AST, and urine creatinine; Model 4 was further adjusted for experiences of heart disease, smoking status, water intake, and caffeine intake.

## 3. Results

### 3.1. Characteristics of the Study Population 

The demographic information of the eligible subjects in the study is shown in Table 1. The mean age of the participants was 47.7 ± 17.79 years old, and 49.8% of participants were male and 43.5% were ever-smokers. Median of baseline variables is as follows: BMI 28.89 kg/m^2^, AST 25.54 U/L, uCr 0.89 mg/dL, and FPG 102.92 mg/dL.

### 3.2. Urinary Caffeine Metabolite Concentrations and Urine Flow Rate 

Associations between urinary caffeine metabolite concentrations and urine flow rate are demonstrated in Table 2. Positive correlations were discovered by linear regression analysis in caffeine and several of its metabolites: 1-methyluric acid (β coefficient = 0.068, *p* < 0.001), 1,7-dimethyluric acid (β coefficient = 0.091, *p* = 0.047), 1,3,7-trimethyluric acid (β coefficient = 1.806, *p* = 0.007), 1-methylxanthine (β coefficient = 0.152, *p* < 0.001), 7-methylxanthine (β coefficient = 0.07, *p* = 0.005), 1,3-dimethylxanthine (theophylline, β coefficient = 1.177, *p* < 0.001), 1,7-dimethylxanthine (paraxanthine, β coefficient = 0.587, *p* < 0.001), 3,7-dimethylxanthine (theobromine, β coefficient = 0.316, *p* < 0.001), 1,3,7-trimethylxanthine (caffeine, β coefficient = 1.102, *p* < 0.001), and 5-acetylamino-6-amino-3-methyluracil (β coefficient = 0.053, *p* = 0.004). Notably, additional adjustments for all covariates did not affect the statistical significance in the aforementioned metabolites. We further categorized our participants into subgroups by gender in Table 3 and by age in Table 4. Paraxanthine (*p* < 0.001 in total population, male, female, under and over 60 years old), theobromine (*p* < 0.001 in total population, male, female, under 60 years old; *p* = 0.011 over 60 years old), and caffeine (*p* < 0.001 in total population, male, female, under and over 60 years old) were the three showing significant positive correlations in all subgroups. 

Table 3 shows the results for male and female subgroups, respectively. In the male subgroup, caffeine and 14 of its metabolites revealed positive correlations with urine flow rate, in contrast with the female subgroup, with a number of 5. Among them, 1-methyluric acid, 7-methyluric acid, 1,7-dimethyluric acid, 3,7-dimethyluric acid, 1,3,7-trimethyluric acid, 3-methylxanthine, 7-methylxanthine, and 5-acetylamino-6-amino-3-methyluracil presented significance in males but not females. 1,3-Dimethylxanthine (theophylline) was the only one showing a positive correlation in females but marginal significance in males (*p*-value = 0.067 in males, *p*-value < 0.001 in females). 

Table 4 shows the results in participants aged under and over 60 years old, respectively, referring to the cutoff point of the elderly population agreed on by the United Nations [20]. In the subgroup of age <60 years old, caffeine and 14 of its metabolites all revealed positive correlations. On the contrary, in the elderly subgroup ≥ 60 years old, the number of metabolites presenting significant correlations shrank to 3, namely, paraxanthine (*p*-value <0.001), theobromine (*p*-value = 0.011), and caffeine (*p*-value <0.001).

## 4. Discussion

In the US population, we found that urine levels of caffeine metabolites were positively associated with urine flow rate. Furthermore, there are more caffeine metabolites showing a flow-dependency in males than females and more in younger participants than older ones. Notably, caffeine and two of its metabolites, paraxanthine and theobromine, revealed significance across all subgroups. They were the main primary metabolites of caffeine, with paraxanthine composing 84% and theobromine composing 12% [18]. Other metabolites were formed by successive demethylations and hydroxylations. The large proportion of caffeine metabolites being flow-dependent across subgroups proves that urine flow rate is a nonnegligible influencing factor in caffeine excretion.

Previous literature discussing caffeine metabolites and urine flow rate focused more on theophylline, which not only belongs to the same methylxanthine family as caffeine but is itself also one of the main urinary metabolites [16,17,21,22,23,24,25]. Previous studies have mentioned the renal clearance of theophylline being highly dependent on urine flow rate [16,17,21,22,23]. Some pharmacologists have developed a mathematical model that explains the dependence of renal clearance on urine flow rate in drugs such as theophylline, ethanol, and butabarbital [24,25]. Being siblings in the same methylxanthine family, with similar metabolic pathways, caffeine and 14 of its metabolites presenting flow-dependency in our study is plausible.

Other studies focusing directly on caffeine were of limited sample size and did not mention comparisons among gender and age, two unneglectable factors that determine urine flow rate. One early study observed a positive association in 10 elderly men [26], while another proposed a positive relationship in 16 volunteers [27]. Several reports mentioned different flow-dependency of caffeine metabolite ratios (MRs) contributing to different roles in the assessment of cytochrome P450 1A2 (CYP1A2) activity [28,29]. These studies revealed evident results but lacked further comparisons in the subgroups. Our study was composed of 1410 participants from a representative sample, providing highly robust evidence with a much bigger number of participants, and made comparisons in subgroups to investigate whether different tendencies existed.

In the present study, we observed differences in the number of caffeine metabolites showing a relationship with urine flow rate. Males outnumbered females, and younger participants outnumbered older ones. We surveyed the factors affecting caffeine metabolism and urine flow rate and found possible reasons for this difference. The rate of caffeine metabolism primarily depends on the genetic variability of enzymes dominating caffeine breakdown, smoking status, alcohol intake, specific medications, liver diseases, and pregnancy [30,31,32]. CYPIA2, the main enzyme metabolizing caffeine, is known to show higher activity in men [33]. In contrast, female hormones decrease CYPIA2 activity during pregnancy and with oral contraceptive use [34]. Furthermore, studies have described that males show a significant decline in urinary flow rate with age, whereas females show less variation in urine flow rate with respect to age [35]. In sum, a slower rate of caffeine metabolism and a more constant feature of urine flow rate in females may explain the observation of less metabolites being flow-dependent in this group. In addition, the urine flow rate declines with age due to various reasons aside from the one that was already excluded in our study, benign prostate hyperplasia. Prolapsed bladder after vaginal childbirth and menopause in females are also factors that may influence urine flow rate [35]. In the elderly group, increased factors affecting the urine flow rate may perturb and weaken the flow-dependency of metabolites.

The potential mechanism that links urinary caffeine metabolites and the urine flow rate together might contribute to the physiological interplay in the kidneys. Tang-Liu et al. established a model that explained drugs with a dependence of renal clearance on urine flow rate [24]. Theophylline and caffeine fall into the category of those that are reabsorbed in the renal tubule, but the diffusional rate is less than that of water. This results in a disequilibrium state that causes the renal clearance of these drugs to be dependent on urine flow rates. Other drugs that showed no flow-dependency were either not reabsorbed at all or their diffusional rate was equal to or greater than that of water. In another study, the rate–concentration curve of theophylline was depicted to be convex-ascending but not linear. The clearance of drugs increases markedly with urine flow up to a certain degree and, thereafter, increases only slightly [18].

The clinical application of assessing caffeine and its metabolites in urine is multifaceted. Medical research fields extend physiology, psychology, and pharmacology. The finding in our study illustrates the necessity of controlling urine flow rates in studies relevant to caffeine, especially when subjects are male and with younger participants. Restriction of fluid intake, salt intake, avoidance of diuretic foods, and adjustment of renal function are reasonable means to approach more adequate assessment. Moreover, other practice could be inferred from this observation. Increasing urine flow rate can act as a means of detoxifying in caffeine-overdosed patients. Those who wish to stay awake by caffeine may consider avoiding factors that speed up their urine flow rate. These practices require further studies to validate their effectiveness. However, they are worth a try according to the flow-dependent feature demonstrated in this study.

Several limitations of this study should be mentioned. Firstly, NHANES is a cross-sectional study in which urinary caffeine metabolites and urine flow rate were examined at one particular time point rather than continuously collected for a long period of time. The causal relationship could not be established due to possible biased results by single measurement. Secondly, we put more emphasis on the dataset from NHANES 2011–2012 but less on NHANES 2009–2010, which was another cycle that collected urinary caffeine metabolite data. The official website mentioned the different instruments used in the two cycles, which may cause analytical bias due to the earlier one requiring the specimens to be analyzed twice. Even so, the results of 2009–2010 of 1853 participants in Table A1 still revealed a similar trend of flow-dependency, only that the amount of metabolites is not the same. Thirdly, we measured the average urine flow rate rather than the peak urine flow rate. Combining both certainly offers a more comprehensive view of urodynamic studies, but peak flow rate requires more complicated calculations with uroflowmetry. Previous comparable studies have utilized the average urine flow rate, as shown in Table 5 [17,21,22,23,25,26,27,28,29], and thus we followed their practice.

## 5. Conclusions

A positive association exists between several urinary caffeine metabolites and urine flow rate. The number of metabolites showing certain flow-dependency is higher in males than females and also higher in young participants compared to elderly participants. Further studies are necessary to elucidate the mechanisms underlying the flow-dependency appearance of caffeine metabolites in urine. Our study highlights the importance of considering the urine flow rate as an influencing factor in interpretations of urinary data regarding caffeine.

## Figures and Tables

**Figure 1 nutrients-12-02803-f001:**
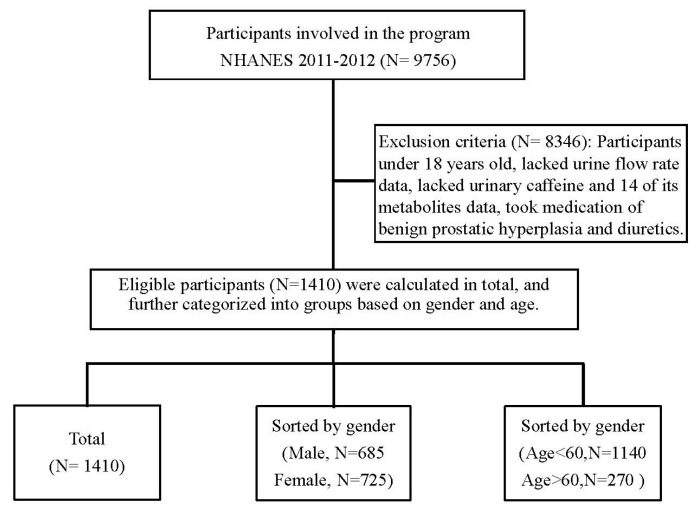
Flow chart of participant recruitment.

**Table 1 nutrients-12-02803-t001:** Characteristics of participants.

Variables	Median (IQR) or Percent (%)
Continuous variables	
Age (years)	47.70 ± 17.79
BMI (kg/m^2^)	28.89 ± 6.85
Aspartate aminotransferase (AST)(U/L)	25.54 ± 14.01
urine creatinine (mg/dL)	0.89 ± 0.29
serum fasting glucose (mg/dL)	102.92 ± 41.51
1-methyluric acid (umol/L)	0.91 ± 1.19
3-methyluric acid (umol/L)	0.01 ± 0.02
7-methyluric acid (umol/L)	0.22 ± 0.34
1,3-dimethyluric acid (umol/L)	0.11 ± 0.26
1,7-dimethyluric acid (umol/L)	0.42 ± 0.49
3,7-dimethyluric acid (umol/L)	0.01 ± 0.02
1,3,7-trimethyluric acid (umol/L)	0.03 ± 0.03
1-methylxanthine (umol/L)	0.48 ± 0.68
3-methylxanthine (umol/L)	0.42 ± 0.61
7-methylxanthine (umol/L)	0.67 ± 0.92
1,3-dimethylxanthine (theophylline) (umol/L)	0.03 ± 0.07
1,7-dimethylxanthine (paraxanthine) (umol/L)	0.29 ± 0.40
3,7-dimethylxanthine (theobromine) (umol/L)	0.28 ± 0.44
1,3,7-trimethylxanthine (caffeine) (umol/L)	0.09 ± 0.18
5-acetylamino-6-amino-3-methyluracil (uM/L)	0.96 ± 1.25
Caffeine intake on the exam day (mg)	142.74 ± 192.73
Total plain water drank the day before exam (mg)	1130.95 ± 1213.59
Categorical variables	
Gender	
Male	49.8
Female	50.2
Race	
Mexican American	9.5
Other Hispanic	10.5
Non-Hispanic White	36.9
Non-Hispanic Black	26.8
Other Race—including Multi-Racial	16.4
Heart disease—ever had a diagnosis	
Congestive heart failure	3.5
Coronary heart disease	4
Angina	2.6
Heart attack	3.8
Smoking	43.5

interquartile range (IQR).

**Table 2 nutrients-12-02803-t002:** Association between urinary caffeine metabolites and urine flow rate.

Variables	Model 1 β(95% CI)	*p* Value	Model 2 β(95% CI)	*p* Value	Model 3 β(95% CI)	*p* Value	Model 4 β(95% CI)	*p* Value
1-methyluric acid	0.072	<0.001	0.074	<0.001	0.083	<0.001	0.055	0.015
	(0.035, 0.110)		(0.037, 0.112)		(0.045, 0.121)		(0.011, 0.099)	
3-methyluric acid	1.472	0.310	2.048	0.163	2.651	0.073	1.281	0.399
	(−1.373, 4.318)		(−0.833, 4.930)		(−0.244, 5.546)		(−1.698, 4.260)	
7-methyluric acid	0.083	0.260	0.114	0.126	0.125	0.092	0.042	0.582
	(−0.061, 0.228)		(−0.032, 0.260)		(−0.021, 0.271)		(−0.108, 0.192)	
1,3-dimethyluric acid	0.048	0.565	0.046	0.585	0.055	0.510	−0.012	0.885
	(−0.116, 0.213)		(−0.118, 0.210)		(−0.109, 0.219)		(−0.177, 0.153)	
1,7-dimethyluric acid	0.108	0.020	0.135	0.005	0.147	0.002	0.054	0.322
	(0.017, 0.200)		(0.041, 0.228)		(0.053, 0.240)		(−0.053, 0.161)	
3,7-dimethyluric acid	2.291	0.052	2.825	0.016	2.720	0.02	1.802	0.126
	(−0.022, 4.604)		(0.527, 5.123)		(0.427, 5.013)		(−0.507, 4.110)	
1,3,7-trimethyluric acid	1.936	0.005	2.400	0.001	2.637	<0.001	1.508	0.046
	(0.598, 3.274)		(1.049, 3.751)		(1.287, 3.988)		(0.029, 2.987)	
1-methylxanthine	0.164	<0.001	0.170	<0.001	0.170	<0.001	0.130	0.001
	(0.100, 0.229)		(0.105, 0.234)		(0.105, 0.235)		(0.056, 0.204)	
3-methylxanthine	0.098	0.024	0.125	0.004	0.120	0.006	0.078	0.081
	(0.013, 0.183)		(0.040, 0.211)		(0.035, 0.206)		(−0.010, 0.165)	
7-methylxanthine	0.080	0.002	0.091	<0.001	0.087	0.001	0.063	0.019
	(0.029, 0.131)		(0.040, 0.141)		(0.035, 0.138)		(0.010, 0.115)	
1,3-dimethylxanthine (theophylline)	1.146	<0.001	1.187	<0.001	1.173	<0.001	0.941	0.002
	(0.549, 1.743)		(0.594, 1.780)		(0.579, 1.766)		(0.343, 1.540)	
1,7-dimethylxanthine (paraxanthine)	0.590	<0.001	0.607	<0.001	0.609	<0.001	0.607	<0.001
	(0.483, 0.697)		(0.500, 0.713)		(0.502, 0.717)		(0.488, 0.725)	
3,7-dimethylxanthine (theobromine)	0.368	<0.001	0.398	<0.001	0.386	<0.001	0.347	<0.001
	(0.256, 0.479)		(0.287, 0.509)		(0.275, 0.498)		(0.235, 0.459)	
1,3,7-trimethylxanthine (caffeine)	1.091	<0.001	1.177	<0.001	1.186	<0.001	1.097	<0.001
	(0.855, 1.327)		(0.942, 1.413)		(0.950, 1.422)		(0.845, 1.348)	
5-acetylamino-6-amino-3-methyluracil	0.061	0.001	0.064	0.001	0.065	<0.001	0.029	0.188
	(0.025, 0.097)		(0.028, 0.100)		(0.029, 0.102)		(−0.014, 0.073)	

Model 1 = unadjusted. Model 2 = Model 1 + age, gender, and race/ethnicity. Model 3 = Model 2 + BMI, serum fasting glucose, aspartate aminotransferase (AST), and urine creatinine. Model 4 = Model 3 + congestive heart failure, coronary heart disease, angina, heart attack, smoking, caffeine intake, and water intake. CI, confidence interval.

**Table 3 nutrients-12-02803-t003:** Association between urinary caffeine metabolites and urine flow rate as categorized by gender.

Variables		Male				Female			
		Model 1	Model 2	Model 3	Model 4	Model 1	Model 2	Model 3	Model 4
1-methyluric acid	β(95% CI)	0.094 (0.046, 0.142)	0.099 (0.050, 0.148)	0.110 (0.061, 0.159)	0.089 (0.028,0.149)	0.051 (−0.006,0.108)	0.050 (−0.007,0.108)	0.053 (−0.006,0.111)	0.026 (−0.039,0.091)
	*p* value	<0.001	<0.001	<0.001	0.004	0.078	0.087	0.078	0.436
3-methyluric acid	β(95% CI)	2.721 (−0.861,6.303)	2.878 (−0.771,6.526)	3.305 (−0.353,6.962)	2.007 (−1.785,5.798)	1.262 (−3.159,5.683)	1.089 (−3.441,5.620)	1.492 (−3.083,6.067)	−0.043 (−4.770,4.685)
	*p* value	0.136	0.122	0.077	0.299	0.575	0.637	0.522	0.986
7-methyluric acid	β(95% CI)	0.203 (−0.005,0.412)	0.214 (0.002,0.426)	0.231 (0.020,0.443)	0.130 (−0.094,0.354)	0.052 (−0.147,0.252)	0.045 (−0.158,0.249)	0.043 (−0.160,0.246)	−0.024 (−0.236,0.187)
	*p* value	0.056	0.048	0.032	0.256	0.607	0.661	0.677	0.823
1,3-dimethyluric acid	β(95% CI)	0.012 (−0.145,0.170)	0.013 (−0.145,0.171)	0.021 (−0.136,0.179)	0.015 (−0.173,0.142)	0.433 (−0.175,1.042)	0.428 (−0.195,1.052)	0.468 (−0.160,1.096)	0.039 (−0.679,0.758)
	*p* value	0.876	0.869	0.789	0.847	0.163	0.178	0.144	0.915
1,7-dimethyluric acid	β(95% CI)	0.254 (0.129,0.378)	0.271 (0.143,0.400)	0.281 (0.153,0.409)	0.205 (0.051,0.359)	0.032 (−0.100,0.163)	0.027 (−0.108,0.162)	0.032 (−0.103,0.167)	−0.061 (−0.212,0.091)
	*p* value	<0.001	<0.001	<0.001	0.009	0.637	0.695	0.644	0.433
3,7-dimethyluric acid	β(95% CI)	3.537 (0.185,6.889)	3.551 (0.192,6.910)	3.487 (0.139,6.836)	2.580 (−0.812,5.973)	2.357 (−0.829,5.542)	2.377 (−0.814,5.568)	2.123 (−1.061,5.308)	1.214 (−2.026,4.453)
	*p* value	0.039	0.038	0.041	0.136	0.147	0.144	0.191	0.462
1,3,7-trimethyluric acid	β(95% CI)	3.807 (1.895,5.720)	4.005 (2.046,5.965)	4.325 (2.365,6.284)	3.194 (0.958,5.431)	1.367 (−0.496,3.229)	1.340 (−0.542,3.222)	1.439 (−0.445,3.323)	0.431 (−1.559,2.482)
	*p* value	<0.001	<0.001	<0.001	0.005	0.150	0.162	0.134	0.654
1-methylxanthine	β(95% CI)	0.218 (0.133,0.303)	0.221 (0.135,0.306)	0.223 (0.137,0.309)	0.190 (0.088,0.292)	0.125 (0.030,0.221)	0.125 (0.029,0.221)	0.119 (0.022,0.217)	0.088 (−0.018,0.194)
	*p* value	<0.001	<0.001	<0.001	<0.001	0.010	0.011	0.017	0.103
3-methylxanthine	β(95% CI)	0.169 (0.048,0.290)	0.173 (0.051,0.295)	0.170 (0.051,0.295)	0.123 (−0.002,0.249)	0.093 (−0.026,0.213)	0.092 (−0.029,0.212)	0.081 (−0.040,0.202)	0.042 (−0.082,0.167)
	*p* value	0.006	0.006	0.007	0.054	0.126	0.135	0.191	0.503
7-methylxanthine	β (95% CI)	0.129 (0.052,0.205)	0.129 (0.052,0.206)	0.129 (0.052,0.207)	0.101 (0.021,0.182)	0.068 (0.000,0.137)	0.069 (0.000,0.137)	0.060 (−0.009,0.130)	0.043 (−0.029,0.115)
	*p* value	0.001	0.001	0.001	0.013	0.050	0.050	0.088	0.238
1,3-dimethylxanthine (theophylline)	β(95% CI)	0.515 (−0.072,1.101)	0.519 (−0.070,1.109)	0.530 (−0.057,1.118)	0.403 (−0.182,0.988)	5.657 (3.490,7.373)	5.696 (3.967,7.425)	5.681 (3.922,7.440)	5.309 (3.376,7.242)
	*p* value	0.085	0.084	0.077	0.177	<0.001	<0.001	<0.001	<0.001
1,7-dimethylxanthine (paraxanthine)	β(95% CI)	0.596 (0.460,0.732)	0.607 (0.470,0.744)	0.602 (0.465,0.740)	0.607 (0.453, 0.760)	0.609 (0.446,0.773)	0.609 (0.445,0.774)	0.610 (0.442,0.777)	0.605 (0.421,0.789)
	*p* value	<0.001	<0.001	<0.001	<0.001	<0.001	<0.001	<0.001	<0.001
3,7-dimethylxanthine (theobromine)	β(95% CI)	0.436 (0.279,0.593)	0.439 (0.282,0.597)	0.425 (0.267,0.583)	0.409 (0.249,0.568)	0.370 (0.213,0.527)	0.371 (0.214,0.529)	0.356 (0.198,0.514)	0.308 (0.147,0.468)
	*p* value	<0.001	<0.001	<0.001	<0.001	<0.001	<0.001	<0.001	<0.001
1,3,7-trimethylxanthine (caffeine)	β(95% CI)	1.496 (1.136,1.856)	1.526 (1.163,1.890)	1.514 (1.152,1.876)	1.429 (1.033,1.826)	0.983 (0.670,1.297)	0.989 (0.672,1.306)	0.988 (0.670,1.307)	0.890 (0.552,1.227)
	*p* value	<0.001	<0.001	<0.001	<0.001	<0.001	<0.001	<0.001	<0.001
5-acetylamino-6-amino-3-methyluracil	β(95% CI)	0.094 (0.051,0.138)	0.098 (0.054,0.143)	0.097 (0.052,0.141)	0.074 (0.019,0.129)	0.023 (−0.034,0.080)	0.022 (−0.037,0.080)	0.023 (−0.037,0.082)	−0.025 (−0.095,0.044)
	*p* value	<0.001	<0.001	<0.001	0.008	0.429	0.468	0.454	0.472

Model 1 = unadjusted. Model 2 = Model 1 + age, gender, and race/ethnicity. Model 3 = Model 2 + BMI, serum fasting glucose, aspartate aminotransferase (AST), and urine creatinine. Model 4 = Model 3 + congestive heart failure, coronary heart disease, angina, heart attack, smoking, caffeine intake, and water intake.

**Table 4 nutrients-12-02803-t004:** Association between urinary caffeine metabolites and urine flow rate as categorized by age.

Variables		Age <60				Age ≥ 60			
		Model 1	Model 2	Model 3	Model 4	Model 1	Model 2	Model 3	Model 4
1-methyluric acid	β(95% CI)	0.102 (0.0538, 0.146)	0.098 (0.053, 0.142)	0.103 (0.058, 0.148)	0.076 (0.025,0.127)	0.004 (−0.068,0.075)	0.001 (−0.070,0.072)	0.006 (−0.066,0.079)	−0.003 (−0.126,0.060)
	*p* value	<0.001	<0.001	<0.001	0.004	0.920	0.987	0.861	0.483
3-methyluric acid	β(95% CI)	3.936 (0.409,7.462)	3.833 (0.299,7.368)	4.190 (0.641,7.740)	2.595 (−1.129,6.318)	−2.656 (−7.598,2.286)	−1.480 (−6.431,3.471)	−1.003 (−6.017,4.011)	−1.733 (−7.035,3.569)
	*p* value	0.029	0.034	0.021	0.172	0.291	0.557	0.694	0.521
7-methyluric acid	β(95% CI)	0.212 (0.033,0.391)	0.212 (0.033,0.391)	0.214 (0.035,0.394)	0.130 (−0.057,0.011)	−0.134 (−0.384,0.115)	−0.063 (−0.313,0.187)	−0.065 (−0.315,0.185)	−0.156 (−0.423,0.112)
	*p* value	0.021	0.021	0.019	0.172	0.290	0.619	0.609	0.253
1,3-dimethyluric acid	β(95% CI)	0.912 (0.428,1.395)	0.894 (0.401,1.387)	0.943 (0.447,1.438)	0.546 (−0.022,1.114)	−0.056 (−0.233,0.121)	−0.080 (−0.256,0.095)	−0.086 (−0.261,0.089)	−0.100 (−0.276,0.077)
	*p* value	<0.001	0.001	<0.001	0.059	0.536	0.369	0.333	0.268
1,7-dimethyluric acid	β(95% CI)	0.220 (0.104,0.337)	0.224 (0.105,0.344)	0.229 (0.110,0.349)	0.132 (−0.003,0.267)	−0.056 (−0.206,0.095)	−0.045 (−0.196,0.106)	−0.044 (−0.196,0.109)	−0.124 (−0.305,0.057)
	*p* value	<0.001	<0.001	<0.001	0.056	0.466	0.554	0.572	0.179
3,7-dimethyluric acid	β(95% CI)	2.883 (0.280,5.485)	3.333 (0.754,5.912)	3.317 (0.736,5.897)	2.314 (−0.299,4.926)	−0.044 (−5.104,5.015)	−0.319 (−4.697,5.335)	−0.350 (−5.344,4.644)	−1.263 (−6.403,3.877)
	*p* value	0.030	0.011	0.012	0.083	0.986	0.900	0.890	0.629
1,3,7-trimethyluric acid	β(95% CI)	2.977 (1.365,4.589)	3.220 (1.587,4.853)	3.366 (1.729,5.002)	2.180 (0.420,3.939)	−0.146 (−2.565,2.273)	−0.058 (−2.482,2.366)	0.294 (−2.149,2.737)	−0.487 (−3.270,2.297)
	*p* value	<0.001	<0.001	<0.001	0.015	0.906	0.962	0.813	0.731
1-methylxanthine	β(95% CI)	0.164 (0.094,0.233)	0.160 (0.090,0.229)	0.164 (0.093,0.235)	0.122 (0.043,0.201)	0.162 (−0.012,0.337)	0.144 (−0.030,0.317)	0.108 (−0.069,0.285)	0.068 (−0.146,0.283)
	*p* value	<0.001	<0.001	<0.001	0.002	0.068	0.105	0.232	0.531
3-methylxanthine	β(95% CI)	0.142 (0.039,0.244)	0.159 (0.056,0.262)	0.159 (0.056,0.263)	0.105 (−0.001,0.0211)	0.009 (−0.146,0.164)	0.036 (−0.118,0.191)	0.004 (−0.151,0.159)	−0.015 (−0.175,0.145)
	*p* value	0.007	0.002	0.003	0.051	0.908	0.643	0.957	0.855
7-methylxanthine	β(95% CI)	0.086 (0.029,0.143)	0.094 (0.038,0.150)	0.093 (0.036,0.150)	0.068 (0.010,0.127)	0.054 (−0.062,0.170)	0.066 (−0.049,0.181)	0.035 (−0.081,0.152)	0.013 (−0.110,0.135)
	*p* value	0.003	0.001	0.001	0.022	0.361	0.258	0.549	0.839
1,3-dimethylxanthine (theophylline)	β(95% CI)	5.900 (4.433,7.367)	6.267 (4.792,7.742)	6.401 (4.911,7.891)	5.823 (4.204,7.442)	0.283 (−0.378,0.943)	0.185 (−0.470,0.841)	0.110 (−0.544,0.764)	0.059 (−0.602,0.721)
	*p* value	<0.001	<0.001	<0.001	<0.001	0.401	0.578	0.741	0.860
1,7-dimethylxanthine (paraxanthine)	β(95% CI)	0.618 (0.487,0.749)	0.627 (0.496,0.759)	0.640 (0.508,0.773)	0.613 (0.469,0.757)	0.536 (0.349,0.724)	0.513 (0.326,0.700)	0.479 (0.287,0.671)	0.532 (0.314,0.750)
	*p* value	<0.001	<0.001	<0.001	<0.001	<0.001	<0.001	<0.001	<0.001
3,7-dimethylxanthine (theobromine)	β(95% CI)	0.376 (0.245,0.507)	0.407 (0.278,0.537)	0.405 (0.275,0.536)	0.358 (0.227,0.488)	0.340 (0.123,0.557)	0.344 (0.130,0.559)	0.289 (0.071,0.507)	0.260 (0.036,0.484)
	*p* value	<0.001	<0.001	<0.001	<0.001	0.002	0.002	0.009	0.023
1,3,7-trimethylxanthine (caffeine)	β(95% CI)	1.319 (1.020,1.618)	1.377 (1.078,1.676)	1.398 (1.097,1.698)	1.265 (0.948,1.582)	0.763 (0.374,1.152)	0.791 (0.405,1.176)	0.749 (0.361,1.137)	0.762 (0.338,1.186)
	*p* value	<0.001	<0.001	<0.001	<0.001	<0.001	<0.001	<0.001	<0.001
5-acetylamino-6-amino-3-methyluracil	β(95% CI)	0.094 (0.048,0.141)	0.088 (0.040,0.135)	0.088 (0.040,0.136)	0.045 (−0.001,0.102)	0.020 (−0.037,0.077)	0.018 (−0.039,0.074)	0.016 (−0.041,0.072)	−0.002 (−0.073,0.068)
	*p* value	<0.001	<0.001	<0.001	0.117	0.486	0.539	0.591	0.946

Model 1 = unadjusted. Model 2 = Model 1 + age, gender, and race/ethnicity. Model 3 = Model 2 + BMI, serum fasting glucose, aspartate aminotransferase (AST), and urine creatinine. Model 4 = Model 3 + congestive heart failure, coronary heart disease, angina, heart attack, smoking, caffeine intake, and water intake.

**Table 5 nutrients-12-02803-t005:** Summary of the literature review findings on the association between urinary caffeine metabolites and urine flow rate.

Study Details	Study Design	Participants	Caffeine Metabolites	Evaluation of Urine Flow Rate	Findings on Urinary Caffeine Metabolites and Urine Flow Rate
**Caffeine**					
Our study	cross-sectional study	N = 1410	Caffeine and 14 of its metabolites	Average flow rate	Positive correlations were shown between several urinary metabolites and urine flow rate. Men showed more correlation than females, and the young (age < 60) showed more correlation than the elderly (age > 60).
Blanchard, J. et al. (1983), Scotland [25]	cross-sectional study	N = 16	Caffeine	Average flow rate	Positive correlation between the renal clearance of both unbound (CLU) and total (CLR) caffeine and the mean urine flow rate.
Trang, J.M. et al. (1985), USA [24]	cross-sectional study	N = 10	Caffeine	Average flow rate	Positive correlations were observed between total body clearance (CL), renal clearance (CL), and nonrenal clearance (CL) and urine flow rate (UFR)
Sinués, B. et al. (1999), Spain [26]	cross-sectional study	N = 125	5 urinary caffeine metabolite ratios (MRs)	Average flow rate	MR1, MR3, and MR4 were the most flow-dependent. MR2 was flow-independent. MR5 was less flow-dependent.
Sinués, B. et al. (2002), Spain [27]	cross-sectional study	N = 152	8 caffeine metabolites and 5 urinary caffeine metabolite ratios (MRs)	Average flow rate	7 caffeine metabolites were flow-dependent. MR1, MR3, and MR4 were flow-dependent. MR2 and MR5 were flow-independent.
**Theophylline**					
Our study	cross-sectional study	N = 1410	Theophylline	Average flow rate	Positive correlations were shown between theophylline and urine flow rate in the female subgroup and the young (age <60) subgroup.
Gerhard Levy. et al. (1976), USA [17]	cross-sectional study	N = 6	Theophylline	Average flow rate	Positive correlation was shown between the renal clearance of theophylline and the urine flow rate.
Tang-Liu, D.D.S. et al. (1982), USA [23]	cross-sectional study	N = 14	Theophylline	Average flow rate	Theophylline renal clearance is highly dependent on urine flow rate and is neither concentration- nor dose-related.
St-Pierre, M.V. et al. (1985), USA [19]	cross-sectional study	N = 8	Theophylline and 3 of its major metabolites	Average flow rate	Renal clearance of metabolites was greater after morning dosing, the time with enhanced urine flow rate.
Bonnacker, I. et al. (1989), Germany [20]	cross-sectional study	N = 10	Theophylline and 3 of its metabolites	Average flow rate	The renal clearance of 1,3-DMU, the main metabolite of theophylline, was found to depend both upon urine flow rate and age.
Agbaba, D. et al., (1990), Yugoslavia [21]	cross-sectional study	N = 22	Theophylline	Average flow rate	The dependence of the renal excretion of theophylline on urine flow rate was found after both IV administration and at steady state.

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
