# Peer review of "Exploring the Association between Urine Caffeine Metabolites and Urine Flow Rate: A Cross-Sectional Study"

_nutrients, 2020, doi:10.3390/nu12092803_

Round 1

Reviewer 1 Report

“Urine levels, serum levels, and metabolite ratios of 46 caffeine act as biomarkers for diseases, targets for drugs and probes for enzyme activity”.

The authors should provide a reference for the statement.

Caffeine is present in cocoa, soft drinks as well as coffee etc. Was the intake of caffeine controlled in amounts? Also, different brands of coffee might cause different absorption and distribution. Was that taken into consideration? It is not clear how the metabolites level was normalized to the caffeine intake. Please provide details.

Conceptually, the calculation of urine flow rate is by dividing the volume of the present urine sample by the time duration between the former urination and the present urine collection, i.e., (total urine volume)/(total time duration).

Please provide more details. Why was water intake not taken into consideration?

model 3 was 131 further adjusted for BMI, serum fasting glucose, AST, urine creatinine

Urine creatinine is often used to account for dilution. What was the purpose in this context?

Four models were provided in each analytic group to adjust relevant covariates. The unadjusted model was numbered one; model 2 was adjusted for age, gender, and race; model 3 was  further adjusted for BMI, serum fasting glucose, AST, urine creatinine; model 4 was further adjusted  for experiences of heart disease and smoking status.

Please provide more details as how the adjustments were made.

Previous literature discussing caffeine metabolite and urine flow rate focused more on 198 theophylline, which not only belongs to the same methylxanthine family with caffeine, but also itself 199 being one of the main urinary metabolites.

Provide reference

Our study composed of 1571 participants from a representative 212 sample provided robust evidence for this positive association and made comparisons in subgroups 213 to investigate whether different tendency existed.

Did these subjects have any conditions that could affect flow rate?

Author Response

August 28, 2020

Dear editor:

Thank you for your encouraging letter concerning our manuscript entitled " Exploring the association between urine caffeine metabolites and urine flow rate: a cross-sectional study " by SH, Wu et al, for publication in Nutrients.

We are extremely grateful to you and the reviewers for the constructive critique of our manuscript. We have responded to each of the comments of the referees on separate sheets and deeply appreciated your suggestions that have led to a significant improvement in this article. All the changes are labeled in red color. Accordingly, we resubmit this article to Nutrients.

We look forward to your prompt reply.

Yours sincerely,

Wei-Liang Chen, M.D. PhD

Division of Geriatric Medicine, Department of Family and Community Medicine Tri-Service General Hospital, National Defense Medical Center, Taipei, Taiwan

Number 325, Section 2, Chang-gong Rd, Nei-Hu District, 114, Taipei, Taiwan.

Tel: +886-2-87923311 ext. 16567 

Fax: +886-2-87927057

Answer to Editor’s and Reviewer's comments

Thank you for your positive comments on this manuscript. The responses to the points raised by the reviewers are listed as follows.

Questions and Comments from Reviewers

Reviewer #1:

  1. “Urine levels, serum levels, and metabolite ratios of 46 caffeine act as biomarkers for diseases, targets for drugs and probes for enzyme activity”.

The authors should provide a reference for the statement.

Response: Thank you for your constructive critique. We added proper references in the introduction (page2, line 46).

Neurologists have demonstrated caffeine and its metabolites as biomarkers for Parkinson disease[1]; Benowitz has listed the pharmacologic actions of caffeine on numerous receptors that make it a therapeutic agent[2]; M.S. Faber et.al described the usefulness of caffeine and its metabolite, theophylline, as assessment probes for CYP1A2 activity[3].

References:

  1. Fujimaki, M.; Saiki, S.; Li, Y.; Kaga, N.; Taka, H.; Hatano, T.; Ishikawa, K.-I.; Oji, Y.; Mori, A.; Okuzumi, A.J.N. Serum caffeine and metabolites are reliable biomarkers of early Parkinson disease. 2018, 90, e404-e411.
  2. Benowitz, N.L.J.A.r.o.m. Clinical pharmacology of caffeine. 1990, 41, 277-288.
  3. Faber, M.S.; Jetter, A.; Fuhr, U.J.B.; pharmacology, c.; toxicology. Assessment of CYP1A2 activity in clinical practice: why, how, and when? 2005, 97, 125-134.

  1. Caffeine is present in cocoa, soft drinks as well as coffee etc.

(1) Was the intake of caffeine controlled in amounts?

(2) Also, different brands of coffee might cause different absorption and distribution. Was that taken into consideration?

(3) It is not clear how the metabolites level was normalized to the caffeine intake. Please provide details.

Response: Thank you for your constructive critique.

(1) (2) We searched for related information required. In NHANES 2011-2012, intake of caffeine was measured in Sl unit, mg, in the dietary interview documentation. Potential bias caused by different brands is eliminated because the measurement was presented with Sl units rather than ambiguous descriptions such as the amount of cups per day.

(3) We followed your recommendation and took these factors into consideration by adding both caffeine and water intake into variables for adjustment in model 4. We redid the data analysis and found the characteristic of flow-dependency still existed in urinary caffeine metabolites. All tables were renewed with the updated model 4. After proper normalization, the association between urinary caffeine metabolites and urine flow rate is firm because these potential influencing factors did not alter our previous findings.

  1. Conceptually, the calculation of urine flow rate is by dividing the volume of the present urine sample by the time duration between the former urination and the present urine collection, i.e., (total urine volume)/(total time duration).

Please provide more details. Why was water intake not taken into consideration?

Response: Thank you for your constructive critique. In NHANES 2011-2012, water intake of each participant was documented in the dietary interview. In addition, the laboratory methodology of urine flow rate was clearly stated in the procedure manuals. A figure was provided for depiction of how collection of urine and time records were made, and detailed information were simplified as the description in our context. The following lines excerpted the details provided on the NHANES website.

“The date and time of last urine void, along with the volume of the urine specimen and the date and time of urine collection captured during urine collection will provide National Center for Health Statistics (NCHS) with a measurement of urine flow rate. These items are captured during the SP’s visit to the MEC. All SPs aged 6+ will be eligible for urine flow rate. Urine specimen volume and date and time of collection will be captured only on SPs who submit a urine specimen.

When the SP arrives at the MEC, the coordinator will prompt the SP to write down the time of his or her last urine void on the verification form. The coordinator will record this time. If there is any discrepancy between the time on the reminder letter and the time reported at the MEC, the coordinator should use the time reported at the MEC. Participants will then be asked to void at the MEC where the time of collection and volume of the urine will be recorded. The volume of the urine sample collected at the MEC will be measured and the urine flow rate will be calculated from this information. Up to three voids were collected for the purpose of ensuring sufficient total volume for various analyses, with volumes and timing recorded. Collected samples were composited then aliquoted into separate vessels such that all analyses can be conducted on the composite sample.

The urine flow rate is calculated by dividing the volume of the urine sample collected, by the time duration between the previous urine void and the urine sample collection in the MEC. There may be a maximum of 3 urine flow rates associated with each urine void for a participant, but that depends on the total number of spot urines collected in the MEC.

The Mettler Toledo scale is used for urine collection. Place the urine collection cup on the Mettler Toledo scale. Cover the scale with the clear box cover to protect the scale from draft. Once the stability indicator goes away, the weight result is valid. Scan the barcode on the cup using the barcode scanner next to the scale. Once the urine cup has been successfully scanned, move it to workstation 2. Once a urine sample has been scanned in, the Urine Collection circle on the heads-up screen for that participant will turn blue and indicates that there is a urine sample in the lab ready to be validated by the laboratory technician.

Then, documenting the urine collection including SP ID, Person Name (last, first), Gender, and Age, Collection instance, Volume (mL) in whole numbers, Collection date/time, Collection status, Comment code, and a checkbox for Verify. The bottom of the screen displays the Total Collected (sum in mL of all urine collections) and Total Required (total mL required to meet the SP’s urine protocol).”

As for how the metabolites level was normalized to the water intake, we adding both caffeine and water intake into variables for adjustment in model 4. We redid the data analysis and found the characteristic of flow-dependency still existed in urinary caffeine metabolites. All tables were renewed with the updated model 4. After proper normalization, the association between urinary caffeine metabolites and urine flow rate is firm because these potential influencing factors did not alter our previous findings.

Table 2. Association between urinary caffeine metabolites and urine flow rate.

Variables

Model 1
β(95% CI)

P value

Model 2
β(95% CI)

P value

Model 3
β(95% CI)

P value

Model 4
β(95% CI)

P value

1-methyluric acid

0.072

<0.001

0.074

<0.001

0.083

<0.001

0.055

0.015

(0.035,0.110)

(0.037,0.112)

(0.045,0.121)

(0.011,0.099)

3-methyluric acid

1.472

0.310

2.048

0.163

2.651

0.073

1.281

0.399

(-1.373,4.318)

(-0.833,4.930)

(-0.244,5.546)

(-1.698,4.260)

7-methyluric acid

0.083

0.260

0.114

0.126

0.125

0.092

0.042

0.582

(-0.061,0.228)

(-0.032,0.260)

(-0.021,0.271)

(-0.108,0.192)

1,3-dimethyluric acid

0.048

0.565

0.046

0.585

0.055

0.510

-0.012

0.885

(-0.116,0.213)

(-0.118,0.210)

(-0.109,0.219)

(-0.177,0.153)

1,7-dimethyluric acid

0.108

0.020

0.135

0.005

0.147

0.002

0.054

0.322

(0.017,0.200)

(0.041,0.228)

(0.053,0.240)

(-0.053,0.161)

3,7-dimethyluric acid

2.291

0.052

2.825

0.016

2.720

0.02

1.802

0.126

(-0.022,4.604)

(0.527,5.123)

(0.427,5.013)

(-0.507,4.110)

1,3,7-trimethyluric acid

1.936

0.005

2.400

0.001

2.637

<0.001

1.508

0.046

(0.598, 3.274)

(1.049, 3.751)

(1.287, 3.988)

(0.029, 2.987)

1-methylxanthine

0.164

<0.001

0.170

<0.001

0.170

<0.001

0.130

0.001

(0.100, 0.229)

(0.105, 0.234)

(0.105, 0.235)

(0.056, 0.204)

3-methylxanthine

0.098

0.024

0.125

0.004

0.120

0.006

0.078

0.081

(0.013, 0.183)

(0.040, 0.211)

(0.035, 0.206)

(-0.010, 0.165)

7-methylxanthine

0.080

0.002

0.091

<0.001

0.087

0.001

0.063

0.019

(0.029, 0.131)

(0.040, 0.141)

(0.035, 0.138)

(0.010, 0.115)

1,3-dimethylxanthine (theophylline)

1.146

<0.001

1.187

<0.001

1.173

<0.001

0.941

0.002

(0.549, 1.743)

(0.594, 1.780)

(0.579, 1.766)

(0.343, 1.540)

1,7-dimethylxanthine (paraxanthine)

0.590

<0.001

0.607

<0.001

0.609

<0.001

0.607

<0.001

(0.483, 0.697)

(0.500, 0.713)

(0.502, 0.717)

(0.488, 0.725)

3,7-dimethylxanthine (theobromine)

0.368

<0.001

0.398

<0.001

0.386

<0.001

0.347

<0.001

(0.256, 0.479)

(0.287, 0.509)

(0.275, 0.498)

(0.235, 0.459)

1,3,7-trimethylxanthine (caffeine)

1.091

<0.001

1.177

<0.001

1.186

<0.001

1.097

<0.001

(0.855, 1.327)

(0.942, 1.413)

(0.950, 1.422)

(0.845, 1.348)

5-acetylamino-6-amino-3-methyluracil

0.061

0.001

0.064

0.001

0.065

<0.001

0.029

0.188

(0.025, 0.097)

(0.028, 0.100)

(0.029, 0.102)

(-0.014, 0.073)

Model 1=unadjusted.

Model 2=Model 1 +age, gender, race/ethnicity.

Model 3=Model 2 +BMI, serum fasting glucose, aspartate aminotransferase (AST), urine creatinine.

Model 4=Model 3 +congestive heart failure, coronary heart disease, angina, heart attack, smoking, caffeine intake, and water intake.

BMI: Body Mass Index

Table 3. Association between urinary caffeine metabolites and urine flow rate categorized by gender.

Variables

Male

Female

Model 1

Model 2

Model 3

Model 4

Model 1

Model 2

Model 3

Model 4

1-methyluric acid

β(95% CI)

0.094
(0.046,0.142)

0.099
(0.050,0.148)

0.110
(0.061,0.159)

0.089
(0.028,0.149)

0.051
(-0.006,0.108)

0.050
(-0.007,0.108)

0.053
(-0.006,0.111)

0.026
(-0.039,0.091)

P value

<0.001

<0.001

<0.001

0.004

0.078

0.087

0.078

0.436

3-methyluric acid

β(95% CI)

2.721
(-0.861,6.303)

2.878
(-0.771,6.526)

3.305
(-0.353,6.962)

2.007
(-1.785,5.798)

1.262
(-3.159,5.683)

1.089
(-3.441,5.620)

1.492
(-3.083,6.067)

-0.043
(-4.770,4.685)

P value

0.136

0.122

0.077

0.299

0.575

0.637

0.522

0.986

7-methyluric acid

β(95% CI)

0.203
(-0.005,0.412)

0.214
(0.002,0.426)

0.231
(0.020,0.443)

0.130
(-0.094,0.354)

0.052
(-0.147,0.252)

0.045
(-0.158,0.249)

0.043
(-0.160,0.246)

-0.024
(-0.236,0.187)

P value

0.056

0.048

0.032

0.256

0.607

0.661

0.677

0.823

1,3-dimethyluric acid

β(95% CI)

0.012
(-0.145,0.170)

0.013
(-0.145,0.171)

0.021
(-0.136,0.179)

0.015
(-0.173,0.142)

0.433
(-0.175,1.042)

0.428
(-0.195,1.052)

0.468
(-0.160,1.096)

0.039
(-0.679,0.758)

P value

0.876

0.869

0.789

0.847

0.163

0.178

0.144

0.915

1,7-dimethyluric acid

β(95% CI)

0.254
(0.129,0.378)

0.271
(0.143,0.400)

0.281
(0.153,0.409)

0.205
(0.051,0.359)

0.032
(-0.100,0.163)

0.027
(-0.108,0.162)

0.032
(-0.103,0.167)

-0.061
(-0.212,0.091)

P value

<0.001

<0.001

<0.001

0.009

0.637

0.695

0.644

0.433

3,7-dimethyluric acid

β(95% CI)

3.537
(0.185,6.889)

3.551
(0.192,6.910)

3.487
(0.139,6.836)

2.580
(-0.812,5.973)

2.357
(-0.829,5.542)

2.377
(-0.814,5.568)

2.123
(-1.061,5.308)

1.214
(-2.026,4.453)

P value

0.039

0.038

0.041

0.136

0.147

0.144

0.191

0.462

1,3,7-trimethyluric acid

β(95% CI)

3.807
(1.895,5.720)

4.005
(2.046,5.965)

4.325
(2.365,6.284)

3.194
(0.958,5.431)

1.367
(-0.496,3.229)

1.340
(-0.542,3.222)

1.439
(-0.445,3.323)

0.431
(-1.559,2.482)

P value

<0.001

<0.001

<0.001

0.005

0.150

0.162

0.134

0.654

1-methylxanthine

β(95% CI)

0.218
(0.133,0.303)

0.221
(0.135,0.306)

0.223
(0.137,0.309)

0.190
(0.088,0.292)

0.125
(0.030,0.221)

0.125
(0.029,0.221)

0.119
(0.022,0.217)

0.088
(-0.018,0.194)

P value

<0.001

<0.001

<0.001

<0.001

0.010

0.011

0.017

0.103

3-methylxanthine

β(95% CI)

0.169
(0.048,0.290)

0.173
(0.051,0.295)

0.170
(0.051,0.295)

0.123
(-0.002,0.249)

0.093
(-0.026,0.213)

0.092
(-0.029,0.212)

0.081
(-0.040,0.202)

0.042
(-0.082,0.167)

P value

0.006

0.006

0.007

0.054

0.126

0.135

0.191

0.503

7-methylxanthine

β(95% CI)

0.129
(0.052,0.205)

0.129
(0.052,0.206)

0.129
(0.052,0.207)

0.101
(0.021,0.182)

0.068
(0.000,0.137)

0.069
(0.000,0.137)

0.060
(-0.009,0.130)

0.043
(-0.029,0.115)

P value

0.001

0.001

0.001

0.013

0.050

0.050

0.088

0.238

1,3-dimethylxanthine (theophylline)

β(95% CI)

0.515
(-0.072,1.101)

0.519
(-0.070,1.109)

0.530
(-0.057,1.118)

0.403
(-0.182,0.988)

5.657
(3.490,7.373)

5.696
(3.967,7.425)

5.681
(3.922,7.440)

5.309
(3.376,7.242)

P value

0.085

0.084

0.077

0.177

<0.001

<0.001

<0.001

<0.001

1,7-dimethylxanthine (paraxanthine)

β(95% CI)

0.596
(0.460,0.732)

0.607
(0.470,0.744)

0.602
(0.465,0.740)

0.607
(0.,0.740)

0.609
(0.446,0.773)

0.609
(0.445,0.774)

0.610
(0.442,0.777)

0.605
(0.421,0.789)

P value

<0.001

<0.001

<0.001

<0.001

<0.001

<0.001

<0.001

<0.001

3,7-dimethylxanthine (theobromine)

β(95% CI)

0.436
(0.279,0.593)

0.439
(0.282,0.597)

0.425
(0.267,0.583)

0.409
(0.249,0.568)

0.370
(0.213,0.527)

0.371
(0.214,0.529)

0.356
(0.198,0.514)

0.308
(0.147,0.468)

P value

<0.001

<0.001

<0.001

<0.001

<0.001

<0.001

<0.001

<0.001

1,3,7-trimethylxanthine (caffeine)

β(95% CI)

1.496
(1.136,1.856)

1.526
(1.163,1.890)

1.514
(1.152,1.876)

1.429
(1.033,1.826)

0.983
(0.670,1.297)

0.989
(0.672,1.306)

0.988
(0.670,1.307)

0.890
(0.552,1.227)

P value

<0.001

<0.001

<0.001

<0.001

<0.001

<0.001

<0.001

<0.001

5-acetylamino-6-amino-3-methyluracil

β(95% CI)

0.094
(0.051,0.138)

0.098
(0.054,0.143)

0.097
(0.052,0.141)

0.074
(0.019,0.129)

0.023
(-0.034,0.080)

0.022
(-0.037,0.080)

0.023
(-0.037,0.082)

-0.025
(-0.095,0.044)

P value

<0.001

<0.001

<0.001

0.008

0.429

0.468

0.454

0.472

Model 1=unadjusted.

Model 2=Model 1 +age, race/ethnicity.

Model 3=Model 2 +BMI, serum fasting glucose, aspartate aminotransferase (AST), urine creatinine.

Model 4=Model 3 +congestive heart failure, coronary heart disease, angina, heart attack, smoking, caffeine intake, and water intake.

BMI: Body Mass Index

Table 4. Association between urinary caffeine metabolites and urine flow rate categorized by age.

Variables

Age<60

Age>60

Model 1

Model 2

Model 3

Model 4

Model 1

Model 2

Model 3

Model 4

1-methyluric acid

β(95% CI)

0.102
(0.0538,0.146)

0.098
(0.053,0.142)

0.103
(0.058,0.148)

0.076
(0.025,0.127)

0.004
(-0.068,0.075)

0.001
(-0.070,0.072)

0.006
(-0.066,0.079)

-0.003
(-0.126,0.060)

P value

<0.001

<0.001

<0.001

0.004

0.920

0.987

0.861

0.483

3-methyluric acid

β(95% CI)

3.936
(0.409,7.462)

3.833
(0.299,7.368)

4.190
(0.641,7.740)

2.595
(-1.129,6.318)

-2.656
(-7.598,2.286)

-1.480
(-6.431,3.471)

-1.003
(-6.017,4.011)

-1.733
(-7.035,3.569)

P value

0.029

0.034

0.021

0.172

0.291

0.557

0.694

0.521

7-methyluric acid

β(95% CI)

0.212
(0.033,0.391)

0.212
(0.033,0.391)

0.214
(0.035,0.394)

0.130
(-0.057,0.011)

-0.134
(-0.384,0.115)

-0.063
(-0.313,0.187)

-0.065
(-0.315,0.185)

-0.156
(-0.423,0.112)

P value

0.021

0.021

0.019

0.172

0.290

0.619

0.609

0.253

1,3-dimethyluric acid

β(95% CI)

0.912
(0.428,1.395)

0.894
(0.401,1.387)

0.943
(0.447,1.438)

0.546
(-0.022,1.114)

-0.056
(-0.233,0.121)

-0.080
(-0.256,0.095)

-0.086
(-0.261,0.089)

-0.100
(-0.276,0.077)

P value

<0.001

0.001

<0.001

0.059

0.536

0.369

0.333

0.268

1,7-dimethyluric acid

β(95% CI)

0.220
(0.104,0.337)

0.224
(0.105,0.344)

0.229
(0.110,0.349)

0.132
(-0.003,0.267)

-0.056
(-0.206,0.095)

-0.045
(-0.196,0.106)

-0.044
(-0.196,0.109)

-0.124
(-0.305,0.057)

P value

<0.001

<0.001

<0.001

0.056

0.466

0.554

0.572

0.179

3,7-dimethyluric acid

β(95% CI)

2.883
(0.280,5.485)

3.333
(0.754,5.912)

3.317
(0.736,5.897)

2.314
(-0.299,4.926)

-0.044
(-5.104,5.015)

-0.319
(-4.697,5.335)

-0.350
(-5.344,4.644)

-1.263
(-6.403,3.877)

P value

0.030

0.011

0.012

0.083

0.986

0.900

0.890

0.629

1,3,7-trimethyluric acid

β(95% CI)

2.977
(1.365,4.589)

3.220
(1.587,4.853)

3.366
(1.729,5.002)

2.180
(0.420,3.939)

-0.146
(-2.565,2.273)

-0.058
(-2.482,2.366)

0.294
(-2.149,2.737)

-0.487
(-3.270,2.297)

P value

<0.001

<0.001

<0.001

0.015

0.906

0.962

0.813

0.731

1-methylxanthine

β(95% CI)

0.164
(0.094,0.233)

0.160
(0.090,0.229)

0.164
(0.093,0.235)

0.122
(0.043,0.201)

0.162
(-0.012,0.337)

0.144
(-0.030,0.317)

0.108
(-0.069,0.285)

0.068
(-0.146,0.283)

P value

<0.001

<0.001

<0.001

0.002

0.068

0.105

0.232

0.531

3-methylxanthine

β(95% CI)

0.142
(0.039,0.244)

0.159
(0.056,0.262)

0.159
(0.056,0.263)

0.105
(-0.001,0.0211)

0.009
(-0.146,0.164)

0.036
(-0.118,0.191)

0.004
(-0.151,0.159)

-0.015
(-0.175,0.145)

P value

0.007

0.002

0.003

0.051

0.908

0.643

0.957

0.855

7-methylxanthine

β(95% CI)

0.086
(0.029,0.143)

0.094
(0.038,0.150)

0.093
(0.036,0.150)

0.068
(0.010,0.127)

0.054
(-0.062,0.170)

0.066
(-0.049,0.181)

0.035
(-0.081,0.152)

0.013
(-0.110,0.135)

P value

0.003

0.001

0.001

0.022

0.361

0.258

0.549

0.839

1,3-dimethylxanthine (theophylline)

β(95% CI)

5.900
(4.433,7.367)

6.267
(4.792,7.742)

6.401
(4.911,7.891)

5.823
(4.204,7.442)

0.283
(-0.378,0.943)

0.185
(-0.470,0.841)

0.110
(-0.544,0.764)

0.059
(-0.602,0.721)

P value

<0.001

<0.001

<0.001

<0.001

0.401

0.578

0.741

0.860

1,7-dimethylxanthine (paraxanthine)

β(95% CI)

0.618
(0.487,0.749)

0.627
(0.496,0.759)

0.640
(0.508,0.773)

0.613
(0.469,0.757)

0.536
(0.349,0.724)

0.513
(0.326,0.700)

0.479
(0.287,0.671)

0.532
(0.314,0.750)

P value

<0.001

<0.001

<0.001

<0.001

<0.001

<0.001

<0.001

<0.001

3,7-dimethylxanthine (theobromine)

β(95% CI)

0.376
(0.245,0.507)

0.407
(0.278,0.537)

0.405
(0.275,0.536)

0.358
(0.227,0.488)

0.340
(0.123,0.557)

0.344
(0.130,0.559)

0.289
(0.071,0.507)

0.260
(0.036,0.484)

P value

<0.001

<0.001

<0.001

<0.001

0.002

0.002

0.009

0.023

1,3,7-trimethylxanthine (caffeine)

β(95% CI)

1.319
(1.020,1.618)

1.377
(1.078,1.676)

1.398
(1.097,1.698)

1.265
(0.948,1.582)

0.763
(0.374,1.152)

0.791
(0.405,1.176)

0.749
(0.361,1.137)

0.762
(0.338,1.186)

P value

<0.001

<0.001

<0.001

<0.001

<0.001

<0.001

<0.001

<0.001

5-acetylamino-6-amino-3-methyluracil

β(95% CI)

0.094
(0.048,0.141)

0.088
(0.040,0.135)

0.088
(0.040,0.136)

0.045
(-0.001,0.102)

0.020
(-0.037,0.077)

0.018
(-0.039,0.074)

0.016
(-0.041,0.072)

-0.002
(-0.073,0.068)

P value

<0.001

<0.001

<0.001

0.117

0.486

0.539

0.591

0.946

Model 1=unadjusted.

Model 2=Model 1 +age, gender, race/ethnicity.

Model 3=Model 2 +BMI, serum fasting glucose, aspartate aminotransferase (AST), urine creatinine.

Model 4=Model 3 +congestive heart failure, coronary heart disease, angina, heart attack, smoking, caffeine intake, and water intake.

BMI: Body Mass Index

  1. model 3 was further adjusted for BMI, serum fasting glucose, AST, urine creatinine

Urine creatinine is often used to account for dilution. What was the purpose in this context?

Response: Thank you for your constructive critique. After comprehensive research, we found the adjustment of urine creatinine a common mean in articles regarding urinary metabolites and biomarkers. K.W.A.Tang et al. discussed normalization of urinary biomarkers to creatinine in clinical practice[1], and concluded that the absolute and normalized values should both be reported to make complete interpretation. K.M.O'Brien also supported creatinine adjustment in evaluations of environmental exposures[2]. Other studies utilizing NHANES on urinary product estimation also adopted creatinine corrections in their analyses [3,4]. Collectively, creatinine corrections better reflect realistic clinical conditions, which is why we put it into one of the covariates for adjustment.

References:

  1. Tang, K.W.A.; Toh, Q.C.; Teo, B.W.J.S.m.j. Normalisation of urinary biomarkers to creatinine for clinical practice and research–when and why. 2015, 56, 7.
  2. O’Brien, K.M.; Upson, K.; Buckley, J.P.J.C.e.h.r. Lipid and creatinine adjustment to evaluate health effects of environmental exposures. 2017, 4, 44-50.
  3. Mage, D.T.; Allen, R.H.; Kodali, A.J.J.o.e.s.; epidemiology, e. Creatinine corrections for estimating children's and adult's pesticide intake doses in equilibrium with urinary pesticide and creatinine concentrations. 2008, 18, 360-368.
  4. Chou, C.-W.; Chen, Y.-Y.; Wang, C.-C.; Kao, T.-W.; Wu, C.-J.; Chen, Y.-J.; Zhou, Y.-C.; Chen, W.-L.J.E.S.; Research, P. Urinary biomarkers of polycyclic aromatic hydrocarbons and the association with hearing threshold shifts in the United States adults. 2020, 27, 562-570.

  1. Four models were provided in each analytic group to adjust relevant covariates. The unadjusted model was numbered one; model 2 was adjusted for age, gender, and race; model 3 was further adjusted for BMI, serum fasting glucose, AST, urine creatinine; model 4 was further adjusted for experiences of heart disease and smoking status.

Please provide more details as how the adjustments were made.

Response: Thank you for your constructive critique. Adjustments in four models were

conducted using covariate regression adjustment means. To explain in detail, by placing the confounding variables in “X” in regression equations, it yields "Y" values which represents expected outcomes that has already taken the influencing factors into consideration. Furthermore, the rationale of choosing confounding variables in each model was based on the common application in previous literature using similar statistical analytic methods. Age, gender, and race were common factors that required adjustment in most studies, and were included in model 2,3, and 4. In model 3 and 4, we further adjusted BMI, serum fasting glucose, ALT, urine creatinine, heart diseases and smoking status because we recognized these factors having potential impact on urinary caffeine level or urine flow rate. BMI was demonstrated as confounding factor in one NHANES article [1]; A study discussing SGLT2 blockade proposed the strong association between blood glucose and urine flow rate [2]; Hepatorenal syndrome depicts how kidney is affected by liver damage due to the accumulation of toxin [3]; The effect of urine creatinine was discussed in question 4; Smoking revealed impact on urine flow rate [4]. Other articles listing similar models were listed in the following [5,6].

References:

  1. Hays, S.M.; Aylward, L.L.; Blount, B.C.J.E.h.p. Variation in urinary flow rates according to demographic characteristics and body mass index in NHANES: potential confounding of associations between health outcomes and urinary biomarker concentrations. 2015, 123, 293-300.
  2. Thomson, S.C.; Rieg, T.; Miracle, C.; Mansoury, H.; Whaley, J.; Vallon, V.; Singh, P.J.A.J.o.P.-R., Integrative; Physiology, C. Acute and chronic effects of SGLT2 blockade on glomerular and tubular function in the early diabetic rat. 2012, 302, R75-R83.
  3. Ginès, P.; Guevara, M.; Arroyo, V.; Rodés, J.J.T.L. Hepatorenal syndrome. 2003, 362, 1819-1827.
  4. Ague, C.J.B.p. Urinary catecholamines, flow rate and tobacco smoking. 1974, 1, 229-236.
  5. Zhou, Y.-C.; Fang, W.-H.; Kao, T.-W.; Wang, C.-C.; Chang, Y.-W.; Peng, T.-C.; Wu, C.-J.; Yang, H.-F.; Chan, J.Y.-H.; Chen, W.-L.J.P.o. Exploring the association between thyroid-stimulating hormone and metabolic syndrome: A large population-based study. 2018, 13, e0199209.
  6. Jhuang, Y.-H.; Kao, T.-W.; Peng, T.-C.; Chen, W.-L.; Li, Y.-W.; Chang, P.-K.; Wu, L.-W.J.H.R. Neutrophil to lymphocyte ratio as predictor for incident hypertension: a 9-year cohort study in Taiwan. 2019, 42, 1209-1214.

  1. Previous literature discussing caffeine metabolite and urine flow rate focused more on theophylline, which not only belongs to the same methylxanthine family with caffeine, but also itself being one of the main urinary metabolites.

Provide reference

Response: Thank you for your constructive critique. We followed your recommendations and added references regarding theophylline studies (page 10, line 204).

  1. Our study composed of 1571 participants from a representative sample provided robust evidence for this positive association and made comparisons in subgroups to investigate whether different tendency existed.

Did these subjects have any conditions that could affect flow rate?

Response: Thank you for your constructive critique. According to the NHANES websites, the purpose of this study was providing extensive information about general health and nutrition status of US population. Therefore, as normal distribution in a general population, severe illness that may pose impact on urine flow rate should take small parts. In table 1, we took gender, races, health conditions into considerations. Furthermore, we excluded those taking medication of benign prostatic hyperplasia and diuretics as these may affect urine flow rate (page 2, line 80-81). We made the exclusion criteria clearer in figure 1.

Last, we are deeply honored by the time and effort you spent in reviewing this manuscript. In reviewing and revising our text, we are motivated to read more and thus learn more from your criticisms.

Reviewer 2 Report

Exploring the association between urine caffeine metabolites and urine flow rate: a cross-sectional study

Objective: establish relationship between urinary metabolites and urinary flow rate

Line 41            It would be more effective to discuss some brief actual findings from the role of caffeine in psychology, sleep disturbance etc.

Overall, this introduction is a bit weak. It would be helpful to expand on the second paragraph as to why this issue is worth investigating. Also there is very little information about what has already been studied in this field, which would be helpful to know going in to the body of this paper.

Line 115          Change “elucidated” to “defined as”

Figure 2 is not very comprehensive – the arrows, especially the jagged one at the top do not really make sense. The use of just blocks and circles as people is too simplistic. <60 and >60 should be defined as age. I didn’t realise that any of the illustrations were supposed to be people until I noticed the glasses on the elderly person. At the very least, this schematic should be explained or defined in the caption.

Line 240          Is urinary flow rate not usually controlled for? Are these examples of studies where it was not controlled that you believe may have a different outcome if it was controlled for? Those studies should be mentioned here.

Line 245          How practical or useful are the “other practices” mentioned? Can you make a statement based on your own data about how decreasing urinary flow would help significantly increase the metabolites associated with caffeine-induced alertness? If you cannot, this statement seems a little sensational.

Line 250          Could the data from 2009-2010 be considered anyways? Not all scientists have the same instruments at their disposal, so it might be interesting to see if the associations made are visible in a different data set where metabolites were measured in a similar way. If the data is taken as a separate step, you could still look for similar trends, which would be a useful contribution or validation of findings for the set that was observed.

Line 256          What is being said here? I am not sure what “exquisite” is meant to mean in this context.

Overall, the science is fine, but the rationale and significance of this work could be vastly improved upon, especially with respect to explanations of how flow rate affecting metabolite excretion may influence findings in clinical studies. Also, hypotheses of why some metabolites are affected and others are not would be important information to include in the discussion. Furthermore, and importantly, the addition of the 2009-2010 data set would help determine the validity of the initial finding from the data presented within the manuscript.

Aside from the science, the English language could use some editing. The grammar is mostly okay- a few missing articles here and there, but word choice in particular makes some of the sentences a bit confusing.

Author Response

August 28, 2020

Dear editor:

Thank you for your encouraging letter concerning our manuscript entitled " Exploring the association between urine caffeine metabolites and urine flow rate: a cross-sectional study " by SH, Wu et al, for publication in Nutrients.

We are extremely grateful to you and the reviewers for the constructive critique of our manuscript. We have responded to each of the comments of the referees on separate sheets and deeply appreciated your suggestions that have led to a significant improvement in this article. All the changes are labeled in red color. Accordingly, we resubmit this article to Nutrients.

We look forward to your prompt reply.

Yours sincerely,

Wei-Liang Chen, M.D. PhD

Division of Geriatric Medicine, Department of Family and Community Medicine Tri-Service General Hospital, National Defense Medical Center, Taipei, Taiwan

Number 325, Section 2, Chang-gong Rd, Nei-Hu District, 114, Taipei, Taiwan.

Tel: +886-2-87923311 ext. 16567 

Fax: +886-2-87927057

Reviewer #2:

  1. Objective: establish relationship between urinary metabolites and urinary flow rate

Line 41 It would be more effective to discuss some brief actual findings from the role of caffeine in psychology, sleep disturbance etc.

Overall, this introduction is a bit weak. It would be helpful to expand on the second paragraph as to why this issue is worth investigating. Also there is very little information about what has already been studied in this field, which would be helpful to know going in to the body of this paper.

Response: Thank you for your constructive critique. We followed your recommendations and expanded a third paragraph in the" introduction” section to explain why this issue is worth investigating. The added context are as following.

Urine flow rate is undoubtedly a crucial factor when interpreting data regarding urinary caffeine metabolism, and thus the association between urinary caffeine metabolite concentrations and urine flow rate worth attention. Previous literature discussing flow-dependency put more focus on theophylline, one of the caffeine metabolites that is well known for its therapeutic effects on asthma and chronic obstructive pulmonary disease(COPD). However, comprehensive studies about other caffeine metabolites is lacking. The purpose of our study was to investigate the relationship between 14 main urinary caffeine metabolites and urine flow rate, expecting to provide more information about the flow-dependent characteristic of caffeine. (page 2, line 48-55)

  1. Line 115 Change “elucidated” to “defined as”

Response: Thank you for your constructive critique. We followed your recommendation and change "elucidated" to "defined as” (page 4, line 122).

  1. Figure 2 is not very comprehensive – the arrows, especially the jagged one at the top do not really make sense. The use of just blocks and circles as people is too simplistic. <60 and >60 should be defined as age. I didn’t realise that any of the illustrations were supposed to be people until I noticed the glasses on the elderly person. At the very least, this schematic should be explained or defined in the caption.

Response: Thank you for your constructive critique. After thorough consideration, we

decided to remove figure 2 due to the over-simplistic presentation that may mislead understanding. Design of this study is detailedly described in materials and methods. Thank you for your comments and we will make clear illustrations when creating similar figures next time.

  1. Line 240 Is urinary flow rate not usually controlled for? Are these examples of studies where it was not controlled that you believe may have a different outcome if it was controlled for? Those studies should be mentioned here.

Response: Thank you for your constructive critique. A study [1] investigating accelerated caffeine metabolism after omeprazole treatment calculated urinary metabolite ratios. Their volunteers were only restricted with intake of methylxanthines such as caffeine or theobromine preceding the study, but no water restrictions. Another study aiming at renal clearance of theophylline excluded children receiving drugs such as erythromycin which alters the pharmacokinetics of theophylline [2]. The above articles showed that designs of these studies mostly excluded substances or medication that may affect their main subjects, but rarely controlled urine flow rate or water intake as they may not be aware that this is an influencing factor. Though we are not sure that whether different outcomes would appear if they controlled urine flow rate, but we are certain that outcomes would be more convincing if they did so.

References:

  1. Rost, K.L.; Roots, I.J.C.P.; Therapeutics. Accelerated caffeine metabolism after omeprazole treatment is indicated by urinary metabolite ratios: coincidence with plasma clearance and breath test. 1994, 55, 402-411.
  2. Bonnacker, I.; Berdel, D.; Süverkrüp, R.; Berg, A.v.J.E.j.o.c.p. Renal clearance of theophylline and its major metabolites: age and urine flow dependency in paediatric patients. 1989, 36, 145-150.

  1. Line 245 How practical or useful are the “other practices” mentioned? Can you make a statement based on your own data about how decreasing urinary flow would help significantly increase the metabolites associated with caffeine-induced alertness? If you cannot, this statement seems a little sensational.

Response: Thank you for your constructive critique. "Other practice” hereby indicates the means of increasing urine flow rate for detoxification and decreasing flow rate for maintaining awareness. The first statement is a reasonable speculation because hydration (drinking more water) is one of the methods suggested when people expect to flush excess caffeine from body [1]. Previous literature also proposed repeated hemodialysis sessions having benefits in caffeine fatal intoxication cases [2]. The latter statement regarding caffeine-induced alertness is of opposite action, and thus we infer such means of decreasing flow rate like avoiding hydration is feasible. Regrettably, there is no evidence at present time supporting our idea. Further studies are required to validate the speculations. We made changes in the statements in page14, line 258-260 to clarify that "other practices” are hypothesis that need to be proved.

References:

  1. Lindberg, S. How to Flush the Body of Caffeine. Availabe online: https://www.livestrong.com/article/349564-how-to-flush-the-body-of-caffeine/ (accessed on Aug 26).
  2. Fausch, K.; Uehlinger, D.E.; Jakob, S.; Pasch, A.J.C.k.j. Haemodialysis in massive caffeine intoxication. 2012, 5, 150-152.

  1. Line 250 Could the data from 2009-2010 be considered anyways? Not all scientists have the same instruments at their disposal, so it might be interesting to see if the associations made are visible in a different data set where metabolites were measured in a similar way. If the data is taken as a separate step, you could still look for similar trends, which would be a useful contribution or validation of findings for the set that was observed.

Response: Thank you for your constructive critique. We followed your recommendation and analyzed data from 2009-2010 even though the instruments used were different from 2011-2012. Same analytic methods were performed, and we constructed the following table. As we can see, though the amount of caffeine metabolites revealing flow-dependency was not the same, but still ** showed the characteristic. Thank you for the advice, we did find similar trends in a different data set which further validate the findings of our study. We will provide this result in the supplement information (page 14, line 268-270).

Supplement Table 1. Association between urinary caffeine metabolites and urine flow rate in NHANES 2009-2010.

Variables

Model 1
β(95% CI)

P value

Model 2
β(95% CI)

P value

Model 3
β(95% CI)

P value

Model 4
β(95% CI)

P value

1-methyluric acid

0.041

0.008

0.049

0.001

0.050

0.001

0.032

0.072

(0.011,0.070)

(0.019,0.079)

(0.020,0.080)

(-0.003,0.066)

3-methyluric acid

-1.087

0.307

0.096

0.930

0.826

0.073

1.281

0.399

(-3.174,1.000)

(-2.045,2.236)

(-0.244,3.027)

(-1.698,2.282)

7-methyluric acid

-0.002

0.968

0.061

0.287

0.072

0.215

0.025

0.670

(-0.112,0.228)

(-0.052,0.260)

(-0.042,0.271)

(-2.157,0.192)

1,3-dimethyluric acid

0.006

0.851

0.015

0.626

0.014

0.630

0.008

0.783

(-0.064,0.065)

(-0.045,0.074)

(-0.046,0.073)

(-0.051,0.067)

1,7-dimethyluric acid

0.097

0.008

0.136

<0.001

0.137

<0.001

0.088

0.031

(0.025,0.169)

(0.064,0.209)

(0.064,0.209)

(0.008,0.167)

3,7-dimethyluric acid

0.561

0.557

1.367

0.154

1.323

0.169

0.714

0.462

(-1.314,2.435)

(-0.515,3.249)

(-0.561,3.207)

(-1.191,2.618)

1,3,7-trimethyluric acid

2.345

<0.001

2.759

<0.001

2.770

<0.001

2.373

<0.001

(1.405, 3.285)

(1.819, 3.698)

(1.832, 3.709)

(1.383, 3.363)

1-methylxanthine

0.124

<0.001

0.131

<0.001

0.127

<0.001

0.101

0.002

(0.067, 0.180)

(0.075, 0.187)

(0.071, 0.184)

(0.037, 0.166)

3-methylxanthine

0.022

0.526

0.064

0.071

0.063

0.073

0.038

0.280

(-0.043, 0.090)

(-0.005, 0.132)

(-0.006, 0.132)

(-0.016, 0.108)

7-methylxanthine

0.037

0.125

0.052

0.028

0.050

0.037

0.032

0.189

(-0.010, 0.083)

(0.006, 0.099)

(0.003, 0.097)

(0.010, 0.081)

1,3-dimethylxanthine (theophylline)

0.897

<0.001

1.004

<0.001

0.986

<0.001

0.896

0.002

(0.494, 1.300)

(0.604, 1.405)

(0.586, 1.387)

(0.496, 1.296)

1,7-dimethylxanthine (paraxanthine)

0.538

<0.001

0.565

<0.001

0.558

<0.001

0.552

<0.001

(0.446, 0.631)

(0.473, 0.656)

(0.466, 0.650)

(0.454, 0.650)

3,7-dimethylxanthine (theobromine)

0.358

<0.001

0.409

<0.001

0.403

<0.001

0.382

<0.001

(0.271, 0.445)

(0.323, 0.496)

(0.316, 0.490)

(0.295, 0.469)

1,3,7-trimethylxanthine (caffeine)

1.079

<0.001

1.153

<0.001

1.153

<0.001

1.097

<0.001

(0.910, 1.248)

(0.985, 1.321)

(0.985, 1.321)

(0.924, 1.270)

5-acetylamino-6-amino-3-methyluracil

0.036

0.031

0.046

0.006

0.045

0.006

0.024

0.213

(0.003, 0.068)

(0.013, 0.078)

(0.013, 0.078)

(-0.014, 0.061)

Model 1=unadjusted.

Model 2=Model 1 +age, gender, race/ethnicity.

Model 3=Model 2 +BMI, serum fasting glucose, aspartate aminotransferase (AST), urine creatinine.

Model 4=Model 3 +congestive heart failure, coronary heart disease, angina, heart attack, smoking, caffeine intake, and water intake.

BMI: Body Mass Index

  1. Line 256 What is being said here? I am not sure what “exquisite” is meant to mean in this context.

Response: Thank you for your constructive critique. The word “exquisite” is changed to "complicated” to make better explanation. We mean that peak urine flow rate is more difficult in calculation because it requires uroflowmetry (page14, line 272).

Last, we are deeply honored by the time and effort you spent in reviewing this manuscript. In reviewing and revising our text, we are motivated to read more and thus learn more from your criticisms.
